# A Novel Method of Si and Si_3_N_4_ Powder Resources Recycling: Cold Bonding Briquettes

**DOI:** 10.3390/ma15165496

**Published:** 2022-08-10

**Authors:** Yuandong Xiong, Ying Li, Huiting Chen, Dejin Qiu, Shiyu Wei, Henrik Saxén, Yaowei Yu

**Affiliations:** 1State Key Laboratory of Advanced Special Steel, Shanghai Key Laboratory of Advanced Ferrometallurgy, School of Materials Science and Engineering, Shanghai University, 99 Shangda Road, Shanghai 200444, China; 2Process and Systems Engineering Laboratory, Faculty of Science and Engineering, Åbo Akademi University, Henriksgatan 2, 20500 Åbo, Finland

**Keywords:** silicon, silicon nitride, cold bonding, compressive strength, high-temperature properties

## Abstract

Silicon nitride (Si_3_N_4_) and silicon powder (Si) are two kinds of harmful solid waste in industrial production. As an environmental and low-consumption method, the cold-bonding technique is a novel method to utilize the problem of powder resource cycling. In this experiment, mechanical and high-temperature properties of Si and Si_3_N_4_ briquettes were studied after cold bonding. The results are as follows: (1) The compressive strength of the Si and Si_3_N_4_ briquettes increased with the improvement of molding pressure. With the same binder (1 wt.%) and water (10 wt.%) addition, the compressive strength of the Si_3_N_4_ briquette arrived at 12,023.53 N under 40 Mpa molding pressure, which is much higher than that of the Si briquette (942.40 N). The Si particles are uneven and irregular, which leads to an intense arch bridge effect in the Si briquette and the compressive strength decrease. Compared with Si powder, the particle size and shape of Si_3_N_4_ is small, uniform, and regular. The influence of the arch bridge effect is smaller than that in the Si briquette. (2) After being treated at 1473 K for 1 h, the compressive strength of the Si briquette increased to 5049.83 N, and the compressive strength of the Si_3_N_4_ briquette had a slight change. The surface of the briquettes was contacted with oxygen and reacted to form an outer shell which mainly contains SiO_2_ in the high-temperature treatment. FT-IR results have shown there were no extra impurities in cold-bonded briquettes when using the organic binder. (3) The microstructure of the cross section of the Si and Si_3_N_4_ briquettes after high-temperature treatment presented that oxygen entered the briquette through the pores and continued to react with the Si and Si_3_N_4_. The outer shell of the Si briquette grew and thickened continuously with the oxygen spreading in the Si briquette. However, because of the smaller particle size and regular shape, little oxygen diffused in the Si_3_N_4_ briquette. The outer shell of the Si_3_N_4_ briquette is fairly thin, so the compressive strength did not change too much.

## 1. Introduction

As a high-quality material, silicon nitride (Si_3_N_4_) is finding applications in ceramics, metallurgy, aircraft, and other industries due to its high-temperature stability, thermal shock resistance, hardness, and wear resistance [1,2,3]. In a reactor, industrial silicon powder is usually used as a raw material to produce Si_3_N_4_ with nitrogen gas through the reaction 3Si(s) + 2N_2_(g) = Si_3_N_4_(s) [4,5]. Compared with industrial silicon powder, silicon nitride has preferable chemical stability and physical properties in applications [2]. However, in the production methods [6,7,8,9] of silicon nitride materials, Si_3_N_4_ powder is generated as a by-product by crushing the lump. Meanwhile, the application of industrial silicon powder, such as the production of Si-Fe alloys [10,11,12] by using a submerged arc furnace, has a considerable powder emission with off-gas. Both Si and Si_3_N_4_ powders collected from industry production are hardly utilized due to the impurity and small particle size. Furthermore, Si and Si_3_N_4_ powders are seriously harmful to the environment if they are simply dumped in the sandpile of the plant. However, they are both good resources and can be used in submerged arc furnaces and downstream industry. Thus, it is meaningful to figure out a way to reuse these harmful powders.

Recently, the cold-bonding technique has been considered as a potential method to treat powder materials [13,14,15], and its merits are in environmental production and low consumption [16]. It is generally used in various fields such as solid waste utilization [17], coal briquette making [18], and metallurgy engineering [19]. Due to the simple flow [20], the importance of the cold-bonding technique development has been the focus of increased attention. Powders, binders, and water are mixed, and the mixture is then molded into wet briquettes under various pressures. At last, the briquettes achieve a certain compressive strength after drying. It is important to note that binder selection [21] and molding pressure [22,23] affect the compressive strength of briquette significantly. Usually, binders are divided into organic, inorganic, and composite binders [21], such as pitch, molasses, starch, bentonite, sodium silicate, and cement, or a mixture of these are used to be binders [24,25,26]. Therefore, it should be noted that both inorganic binder and composite binder used in the cold-bonding process may add more impurity to briquettes even though the compressive strength increased under high temperature [27]. In contrast, using an organic binder can provide excellent compressive strength to the briquette at room temperature. However, the compressive strength of the briquette decreases quickly due to the decomposition of organic binder under high temperatures [16].

Above all, the cold-bonding technique can be used to recycle Si and Si_3_N_4_ powder effectively. However, from another point of view, the quality of material determines the optimal compressive strength of briquettes after being molded. In other words, the difference in chemical and physical properties between Si and Si_3_N_4_ powder may lead to the discrepancy in their briquettes. Therefore, this thesis researches the influence of molding pressure on the compressive strength of cold-bonded briquettes which are made of the two materials by using the same organic binder and water addition. Considering that both Si and Si_3_N_4_ powder are usually applied at high temperatures, the high-temperature properties of their briquettes are studied as well.

## 2. Experimental

### 2.1. Materials

The Si and Si_3_N_4_ powders in this work came from the Inner Mongolia Autonomous region, China. The chemical compositions of the two materials are shown in Table 1. It can be found that the main elements of Si powder are Si and Fe, whose contents are 61.34 wt.% and 20.59 wt.%, respectively. Meanwhile, the Si_3_N_4_ powder contains 38.51 wt.% Si, 23.32 wt.% Fe, 16.05 wt.% N, and 16.83 wt.% Mn, primarily. The purities of Si and Si_3_N_4_ powders are too low to be utilized.

### 2.2. Cold Bonding Briquettes Process

Figure 1 is the schematic of the cold-bonding molding process. Si or Si_3_N_4_ powders (1 kg), water (0.1 kg), and an organic binder (0.01 kg) were mixed sufficiently into a wet mixture at first. Then the wet mixtures were filled into a cylindrical mold and molded into a wet cylindrical briquette under different pressures (5, 10, 20, 30,40 Mpa). Finally, the wet briquettes were dried in a muffle furnace at 180 °C for 12 h in the atmosphere. After the briquettes cooled down to room temperature, the compressive strength of the Si and Si_3_N_4_ briquettes was tested on a universal testing machine. The remainder was heated at 1473 K for 1 h in a muffle furnace, and the compressive strength was tested.

### 2.3. Analysis Method

In this study, the compressive strength was measured on a universal testing machine (UTM, C45.305, MTS, Eden Prairie, MN, USA). X-ray fluorescence spectrometer (XRF-1800, Shimadzu, Japan) was used to test the content of the main elements in the powders. Phase composition of powders was performed on an X-ray diffractometer (Bruker D8 Advance, Bruker, Germany) with a scan step of 10 (°)/min, in the 2θ range from 5° to 90°. The microstructure and elementals distribution were analyzed by using a scanning electron microscopy with an energy dispersive X-ray spectrometer (Hitachi SU-1500, Hitachi, Japan). The briquette was measured on a Fourier transform infrared spectrometer (Thermo Scientific Nicolet iS20, Thermo Fisher Scientific, Waltham, MA, USA) using the KBr disc technique, and the spectra were measured with a resolution of 4 cm−1 between a wave number range of 4000 to 400 cm^−1^.

## 3. Results and Discussion

### 3.1. Effect of Molding Pressure on Compressive Strength of Cold-Bonded Briquette

The compressive strength of different briquettes under various pressures is shown in Table 2. The compressive strength of Si and Si_3_N_4_ briquettes increases with the molding pressure augment. Meanwhile, their compressive strength grows significantly after the molding pressure is over 30 Mpa. When the molding pressure arrives at 40 Mpa, the compressive strength of Si and Si_3_N_4_ briquettes are 942.40 N and 12,023.53 N, respectively. Additionally, the compressive strength of the Si_3_N_4_ briquette is dramatically higher than that of the Si briquette under various molding pressures.

In the cold-bonding process, particles with inhomogeneous size and irregular shape may cause more pores to form inside of the briquette. This phenomenon is named the arch bridge effect [28]. The microstructures of Si and Si_3_N_4_ powders are shown in Figure 2a,b, respectively. Compared with the microstructure of Si_3_N_4_ powder, the particle sizes and shapes of Si powder are uneven and irregular, which means an intense arch bridge effect in the Si briquette after cold bonding. Thus, the compressive strength of the Si briquette is much lower than that of the Si_3_N_4_ briquette.

The compressive-displacement curves of briquettes under different molding pressures are shown in Figure 3a,b, respectively. All of the briquettes have strain-softening characteristics after the compressive strength reaches its maximum. With the molding pressure increasing, pores form due to the decreasing of the arch bridge effect, and the slope of the compressive-strength curves elevates obviously. Matin et al. [29] has studied the process of typical stress versus strain for hard rocks. Similarly, the macroscopic failure process of two briquettes contains four stages: (1) crack closure, (2) elastic region, (3)stable crack growth, and (4) initiation of macro-scale shear failure. At the first stage, the influence of the arch bridge effect reduces gradually, and the slope of the curves increases slowly. When the macroscopic failure process goes to the second stage, the slope elevates to maximum immediately. Meanwhile, the influence of van der Waals force and friction between particles on the compressive strength of the briquettes also increases rapidly. At the third stage, more and more cracks appear inside of briquette, and the slope of compressive-strength curves decreases slightly. Finally, the compressive strength decreases while the slope of curves become negative at the last stage.

Furthermore, it can be seen that the compressive-strength curves of Si briquettes are smoother than that of Si_3_N_4_ briquettes. That is because the small particle size and regular shape of Si_3_N_4_ powder make the friction between particles greater. Additionally, the excellent mechanical properties of Si_3_N_4_ particles [30] also enable the briquette to withstand significant pressure after the arch bridge effect disappears at the second stage.

### 3.2. High-Temperature Properties of Si and Si_3_N_4_ Briquettes

In the production of alloy and ceramics, the Si and Si_3_N_4_ briquettes are possibly used in high-temperature conditions, respectively. Thus, the briquettes molded under 40 Mpa pressure were treated at 1473 K for 1 h by muffle furnace in air. The cross-sectional photographs of Si and Si_3_N_4_ briquettes before and after being treated are shown in Figure 4a,b and Figure 4c,d, respectively. It can be seen from Figure 4a,b clearly that there is a dramatic outer shell that appeared outside of the Si briquette after the high-temperature treatment. As shown in Figure 4c,d, an outer shell is also generated in the high-temperature treatment of the Si_3_N_4_ briquette. However, the outer shell of the Si_3_N_4_ briquette after treatment is too thin to be seen clearly.

Figure 5 indicates the compressive-strength curves of Si and Si_3_N_4_ briquettes before and after being treated at high temperatures. As shown in Figure 5a, the compressive-displacement curves of the Si briquette before and after being treated at 1473 K for 1 h were fairly different. On the one hand, the compressive strength of Si briquette enhanced visibly from 942.40 N to 5049.83 N after being treated. On the other hand, the slope of the compressive-strength curve increased significantly in the first three stages of the macroscopic failure process. The fragmentation characteristics of the briquette changed from strain softening to strain hardening. The first peak of the compressive-strength curve of the Si briquette treated appeared at a displacement of the curve at 2.15 mm; then the compressive strength increased to 5049.83 N.

Compared with the Si briquette, the compressive-strength curve of the Si_3_N_4_ briquette after being treated has only slightly changed, as shown in Figure 5b. A minor peak appeared at a displacement of the curve at 3 mm, and the compressive strength increased to 12,277.91 N subsequently. The compressive strength of the Si_3_N_4_ briquette increased 254.38 N after high-temperature treatment. Meanwhile, the fragmentation characteristics of compressive-strength curves before and after high-temperature treatment were similar.

Figure 6a shows the XRD patterns of the Si briquette and the outer shell. The phase composition of the Si briquette is significantly different from the outer shell. The Si briquette mainly consists of Si and FeSi_2_ (orthorhombic), and the peaks of SiO_2_ and FeSi_2_ (tetragonal) appear in the XRD pattern of the outer shell. In the process of high-temperature treatment, Si and O_2_ react to form SiO_2_, and FeSi_2_ (orthorhombic) transforms to FeSi_2_ (tetragonal). Similarly, as Figure 6b shows, the phase compositions of Si_3_N_4_ powder are α-Si_3_N_4_, β-Si_3_N_4_, MnSiN_2_, and Fe_0_._905_Si_0_._095_. Visibly, the SiO_2_ also appears in the outer shell of the Si_3_N_4_ briquette. Both XRD patterns indicated that the formation of SiO_2_ is the main reason for improving the compressive strength of these two cold-bonded briquettes.

In the high-temperature process, both Si powder and Si_3_N_4_ powder on the briquette surface react with O_2_ and generate SiO_2_ easily as follows reactions (1) and (2):Si + O_2_ → SiO_2_(1)
Si_3_N_4_ + O_2_ → SiO_2_ + NO_2_(2)

As shown in Figure 7, the SiO_2_ produced by the above reactions has undergone a process from amorphous to crystalline and makes the surface of briquettes more compact, and the compressive strength of the briquette improves with the growth of the shell thickness. Meanwhile, the growth of the outer shell stops since amorphous SiO_2_ blocks the pores on the surface of the briquette. Due to the particle sizes and shapes of Si particles being more uneven and irregular than that of Si_3_N_4_ particles, the size of pores on the surface of Si briquettes is too large and the O_2_ diffuses in the briquette more deeply. Thus, the outer shell of the Si briquette is thicker than that of the Si_3_N_4_ briquette, which means the compressive strength of the Si briquette improved more after high-temperature treatment.

### 3.3. FT-IR Analysis of Si and Si_3_N_4_ Briquettes

Organic binders are usually used in the cold-bonding process. However, the briquette has poor performance in high temperatures when using organic binders [25]. Therefore, it is interesting that the compressive strength of the briquette increased after being treated at 1473 K for 1 h. In this research, FT-IR analysis is used to detect the nature of the functional groups [31] present in the powders, briquettes before being treated, and briquettes after being treated.

The results of FT-IR analysis of Si and Si_3_N_4_ (powder, briquette, and outer shell) are shown in Figure 8a,b, respectively. As shown in Figure 8a, the result of the Si powder is similar to that of the Si briquette, which indicates that the organic binder has decomposed at 180 °C. The peak at approximately 3417 cm^−1^ and 1627 cm^−1^ is attributed to the stretching vibration of O-H of adsorbed water and bending vibrations of the crystal water in Si powder [32]. Meanwhile, the O-H band is also existent in the Si briquette and outer shell because of the adsorption of water molecules [33]. Peaks at 1084.08 cm^−1^ and 1037.73 cm^−1^ correspond to the asymmetric stretching vibration of Si-O-Si [34,35], and the appearance of a low-frequency band at 436.51 cm^−1^ and 437.32 cm^−1^ is due to Fe stretching vibration [36]. The FT-IR spectra of the outer shell are quite different from that of the Si powder and briquette. In the spectra of the outer shell, two strong bands appeared at 907.00 cm^−1^ and 960.34 cm^−1^ because of the asymmetric stretching vibration of Si-O-Si and Si-O [37]. Furthermore, a weak band at 536.78 cm^−1^ is also due to Fe stretching vibration [38]; the reason for the change of Fe band position is the crystal transition of FeSi_2_.

Figure 8a shows that the FT-IR spectra of Si_3_N_4_ powder, briquette, and outer shell are similar. The peak at approximately 840 cm^−1^ in FT-IR spectra of Si_3_N_4_ powder, briquette, and the outer shell is mainly attributed to the Si-N bond [39]. Meanwhile, the bands that appeared at approximately 570 cm^−1^ and 440 cm^−1^ are characteristic absorption frequencies of the β- Si_3_N_4_ [40]. The peak at 490 cm^−1^ is assigned to the Si-O-Si bending mode owing to a small amount of SiO_2_ in Si_3_N_4_ powder. In the process of desiccation and high-temperature treatment, reaction (2) takes place and more SiO_2_ is generated. Thus, the peak appears at 1046.38 cm^−1^ and 1046.28 cm^−1^ which is in spectra of the briquette and outer shell, respectively.

### 3.4. Microscopic Morphology of Si and Si_3_N_4_ Briquette after High Temperature

The microscopic morphology and EDS image of the cross section of the Si and Si_3_N_4_ briquette after high-temperature treatment are presented in Figure 9. Figure 9a is the microscopic morphology of the cross section of the Si briquette. It can be seen clearly that the outer shell is fairly smooth, which means the surface of the Si briquette becomes more compact after high-temperature treatment. Figure 9c is the EDS image of the outer shell, and it can be found that the outer shell mainly consisted of SiO_2_. In the high-temperature treatment, the SiO_2_ formed in the outer shell has undergone a process from amorphous to crystalline, which will make the surface more compact. Furthermore, because the pores in the Si briquette are too large to be blocked, oxygen can enter in Si briquette easily. Thus, the outer shell will become thicker and thicker during the high-temperature treatment.

Figure 9b shows the microscopic morphology of the cross section of the Si_3_N_4_ briquette after high-temperature treatment, and Figure 9d is the EDS image of the outer shell. As Figure 9b shows, the thickness of the outer shell is not uniform, which is unlike that of the Si briquette. Dramatically, the outer shell of the Si_3_N_4_ briquette is smoother than that of the Si briquette. In addition, it can also be seen that the outer shell of the Si_3_N_4_ briquette mainly consists of SiO_2_ from the EDS image. Due to the particle size and shape of Si_3_N_4_ being small and regular, the formed pores are diminutive during the cold-bonding process, and reaction (2) can make the pores blocked easily. Meanwhile, a smooth outer shell can also prevent oxygen from entering the inside of the Si_3_N_4_ briquette during the high-temperature treatment. Lastly, the oxygen can enter inside and continue to react in some places with large pores on the surface, and the thickness of the shell is uneven. Although the compressive strength of the Si_3_N_4_ briquette does not change much because the outer shell was fairly thin, it still can be seen that a small fracture peak appears in the compressive-strength curve.

## 4. Conclusions

In this work, a novel method of Si and Si_3_N_4_ powder resources recycling has been proposed. As an environmental and low-consumption method, the Si and Si_3_N_4_ powder resources can be effectively reused via the cold-bonding technique. In the present work, the mechanical properties and high-temperature properties of Si and Si_3_N_4_ briquettes are studied. The results are as follows:(1)The compressive strength of Si and Si_3_N_4_ briquettes increases with the improvement of molding pressure. With the same binder (1 wt.%) and water (10 wt.%) addition, the compressive strength of the Si_3_N_4_ briquette arrives at 12023.53 N under 40 Mpa molding pressure, which is much higher than that of the Si briquette (942.40 N). The microstructure of powders shows that the Si particles have uneven particle sizes and irregular shapes. Thus, an intense arch bridge effect appears in the Si briquette, which leads to a decrease in compressive strength. Compared with Si powder, the particle size and shape of Si_3_N_4_ is small and regular, and the influence of the arch bridge effect is smaller than that in the Si briquette.(2)After being treated at 1473 K for 1 h, the compressive strength of the Si briquette increases to 5049.83 N, and the compressive strength of the Si_3_N_4_ briquette has a slight change. The surface of briquettes is contacted with oxygen and reacts to form an outer shell in the high-temperature treatment. The XRD patterns indicated that the outer shell of Si and Si_3_N_4_ briquettes mainly consist of SiO_2_. FT-IR results have shown there are no extra impurities in cold-bonded briquettes. The organic binder decomposed completely after drying the briquette at 180 °C.(3)The microstructure of the cross section of Si and Si_3_N_4_ briquettes after high-temperature treatment presented that oxygen enters the briquette through the pores and continues to react with the Si and Si_3_N_4_. In this process, the outer shell of the Si briquette grows and thickens continuously because the pores in the Si briquette are large enough to spread oxygen, thus the compressive strength improves obviously. The particle size of Si_3_N_4_ is small and the shape is regular, there are fewer small pores on the surface, and they are easily blocked in the high-temperature process. Lastly, little oxygen enters in Si_3_N_4_ briquette and produces a very thin outer shell, so the compressive strength does not change too much.

## Figures and Tables

**Figure 1 materials-15-05496-f001:**
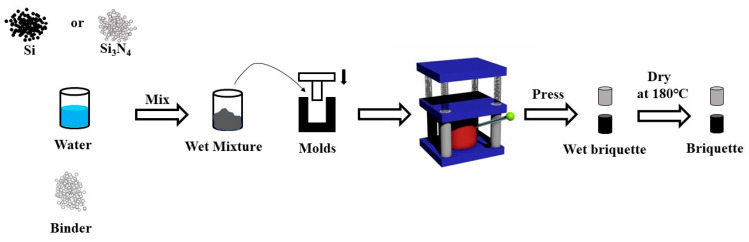
Schematic of the process for making cold-bonded briquettes.

**Figure 2 materials-15-05496-f002:**
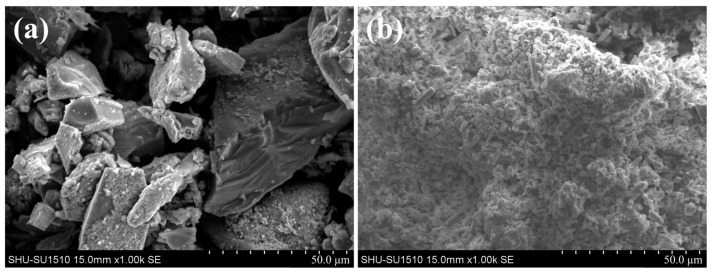
Microstructure of (**a**) Si particles and (**b**) Si_3_N_4_ particles.

**Figure 3 materials-15-05496-f003:**
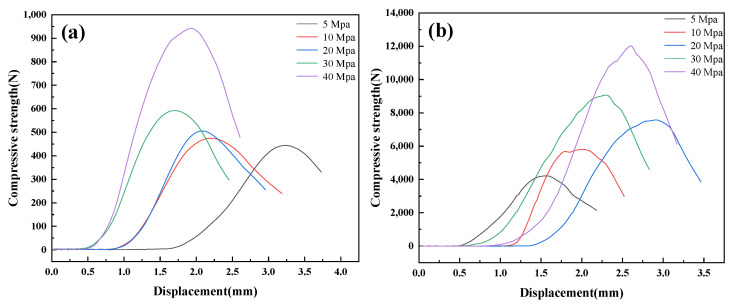
Compressive-displacement curves of the briquettes under various pressures (**a**) Si, (**b**) Si_3_N_4_.

**Figure 4 materials-15-05496-f004:**
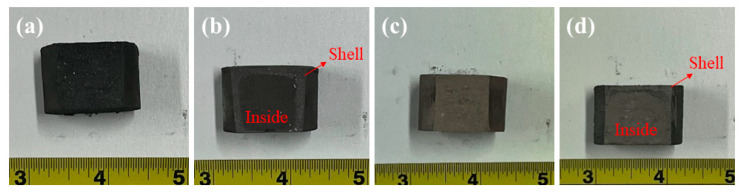
Cross-sectional photographs of (**a**) Si briquette; (**b**) Si briquette after being treated at 1473 K for 1 h; (**c**) Si_3_N_4_ briquette; (**d**) Si_3_N_4_ briquette after being treated at 1473 K for 1 h.

**Figure 5 materials-15-05496-f005:**
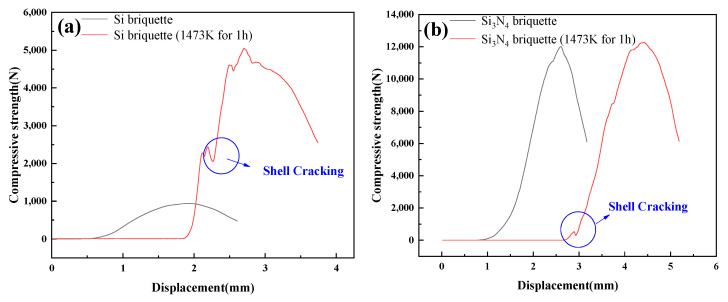
Variation of compressive-displacement curves after being treated at 1473 K for 1 h: (**a**) Si briquette, (**b**) Si_3_N_4_ briquette.

**Figure 6 materials-15-05496-f006:**
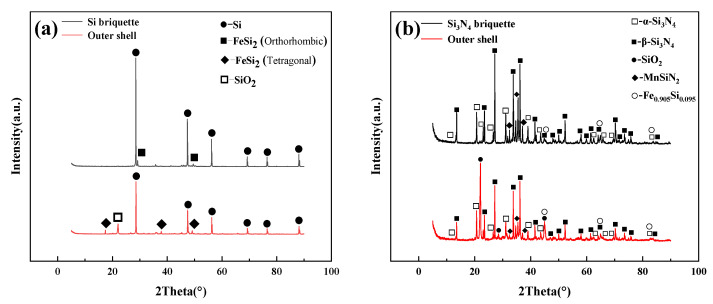
XRD patterns of briquette and outer shell: (**a**) Si; (**b**) Si_3_N_4_.

**Figure 7 materials-15-05496-f007:**
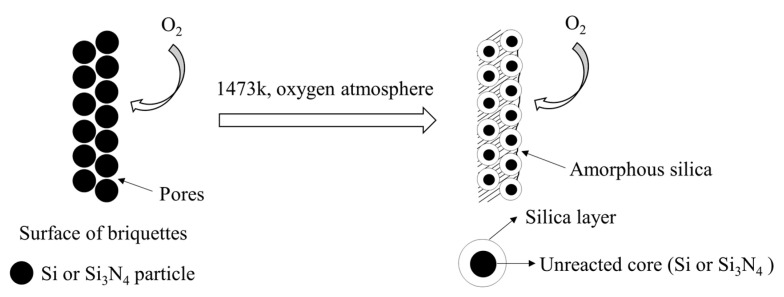
Formation of SiO_2_ layer on the surface of briquettes after heating under an oxygen atmosphere.

**Figure 8 materials-15-05496-f008:**
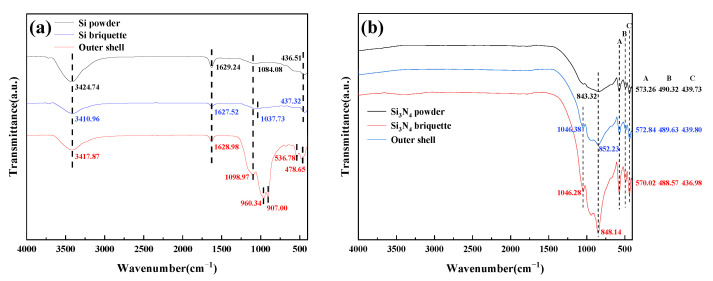
FTIR spectra of powder, briquette, and outer shell (**a**) Si and (**b**) Si_3_N_4_.

**Figure 9 materials-15-05496-f009:**
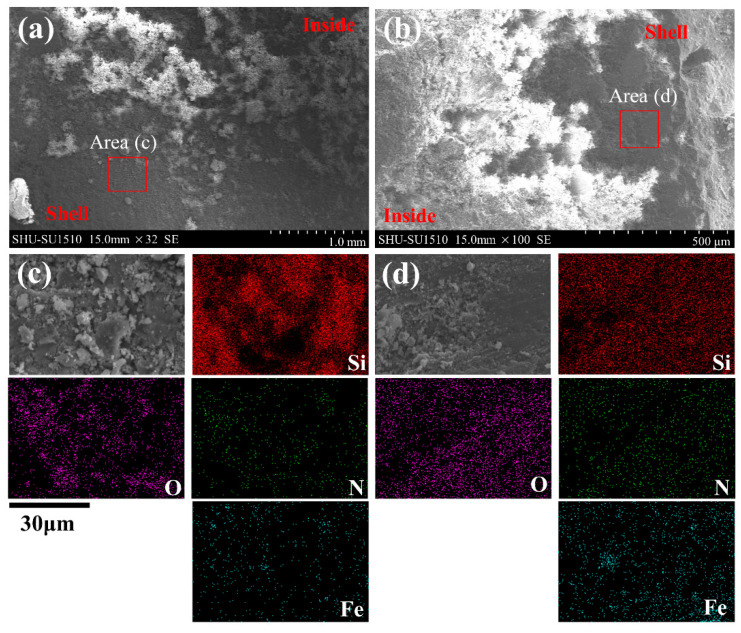
SEM of the cross section of Si and Si_3_N_4_ briquette: (**a**) Si briquette, (**b**) Si_3_N_4_ briquette; EDS image of outer shell: (**c**) Si briquette, (**d**) Si_3_N_4_ briquette.

**Table 1 materials-15-05496-t001:** The chemical compositions of Si and Si_3_N_4_ powders (wt.%).

	Si	Fe	O	Cl	Cu	Ca	Al	N	Ti	Cr	Mn	Sr	Else
Si	61.34	20.59	5.83	3.09	2.71	2.15	0.93	-	0.71	0.29	0.23	0.20	1.90
Si_3_N_4_	38.51	23.32	2.85	-	-	1.24	0.80	16.05	0.12	0.15	16.83	0.14	-

**Table 2 materials-15-05496-t002:** The compressive strength of different briquettes under various pressures.

Molding Pressure/Mpa	Compressive Strength/N	
	Si	Si_3_N_4_
5	444.13	4207.73
10	474.21	5807.40
20	505.35	7579.59
30	592.41	9064.51
40	942.40	12,023.53

## Data Availability

Data are contained within the article and can be requested from the corresponding author.

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
