# Peer review of "A Novel Method of Si and Si3N4 Powder Resources Recycling: Cold Bonding Briquettes"

_materials, 2022, doi:10.3390/ma15165496_

Round 1
Reviewer 1 Report
The article A novel method of Si and Si3N4 powder resources recycling:cold-bonding briquette is devoted to the study of the strength properties of briquettes pressed from Si and Si3N4 for their further utilization. The presented results of the article, to a certain extent, have scientific novelty, but in general, this work is of a practical nature, describing the properties of materials and the influence of external factors such as thermal annealing, leading to oxidation. In general, this article can be accepted for publication after the authors answer a number of questions that arose while reading it.
1. The authors claim that the method of cold pressing proposed by them is one of the inexpensive and cost-effective ones. However, after pressing, the resulting briquettes are also subjected to annealing. The question is, did the authors consider the possibility of hot pressing in order to improve the properties of briquettes?
2. Regarding annealing corrosion, is it related to the fact that annealing occurs in an oxygen-containing environment, and can it be avoided if annealing in an inert atmosphere?
3. In order to avoid the occurrence of softening effects associated with particle size, the authors could consider an additional operation before pressing, such as mechanochemical milling to obtain uniform powder sizes.
4. X-ray diffraction analysis requires additions in the form of a presented diagram with the phase relationship established during the analysis. Also, the authors should explain the presence of the FeO phase in the structure of briquettes; the presence of this phase during thermal annealing can lead to an acceleration of oxidation processes.
5. The EDS results should also include the element Fe, since phases with its presence have been identified.
Author Response
Response to the Reviewer’s Comments
We would like to thank you for your helpful comments. We have followed all the suggestions and made revisions accordingly. The list below gives answers to questions and the details of the changes that we have made in the revised manuscript.
- The authors claim that the method of cold pressing proposed by them is one of the inexpensive and cost-effective ones. However, after pressing, the resulting briquettes are also subjected to annealing. The question is, did the authors consider the possibility of hot pressing in order to improve the properties of briquettes?
Respond: In comparison with hot pressing, cold pressing has a lower consumption. Meantime, the binder is an important factor to influence the properties of briquettes, the adhesive ability of the binder may be weakened by the high temperature in the process of hot pressing. Thus, we did not consider the hot pressing.
- Regarding annealing corrosion, is it related to the fact that annealing occurs in an oxygen-containing environment, and can it be avoided if annealing in an inert atmosphere?
Respond: The corrosion is not only related to the annealing process in an oxygen-containing environment but also the heating and holding process in the same atmosphere. Silicon and silicon nitride will react with oxygen and formate the out-shell, then the properties of briquettes improve. It can be avoided when the whole processes are in an inert atmosphere.
- In order to avoid the occurrence of softening effects associated with particle size, the authors could consider an additional operation before pressing, such as mechanochemical milling to obtain uniform powder sizes.
Respond: It is a good idea that mechanochemical milling to obtain uniform powder sizes. The particle size distribution of materials can affect the properties of cold-pressing briquettes, the studies of the influence of particle size distribution on the properties of silicon and silicon nitride briquettes, and the change of properties after mechanochemical milled are underway. The results will be shown in another paper of our team.
- X-ray diffraction analysis requires additions in the form of a presented diagram with the phase relationship established during the analysis. Also, the authors should explain the presence of the FeO phase in the structure of briquettes; the presence of this phase during thermal annealing can lead to an acceleration of oxidation processes.
Respond: The diagram with the phase relationship established during the analysis has been supplied in Fig.7, the SiO2 produced by the reactions (1) and (2) in different briquettes has undergone a process from amorphous to crystalline and makes the surface of briquettes more compact, and the compressive strength of the briquette improves with the growth of the shell thickness. Meanwhile, the growth of the outer shell stops with amorphous SiO2 blocking the pores of the briquette;
Fig.7 Formation of SiO2 layer on the surface of briquettes after heating under an oxygen atmosphere
Meantime, We are sorry about the analysis mistake of the result of X-ray diffraction in the XRD pattern Fig.6 (b). It has been modified and analyzed carefully again, there is no FeO phase in the Si3N4 briquette in the whole process. the main phase composition of Si3N4 briquettes are α-Si3N4, β-Si3N4, MnSiN2, and Fe0.905Si0.095. The presence of the Fe-containing phase in the structure of briquettes is mainly because the Si3N4 powder is produced by the industrial Si powder in this manuscript which is used to produce FeSi originally. Due to the industrial Si powder containing a certain degree of FeSi2, the Fe-contained phase exists in the Si3N4 powder.
- The EDS results should also include the element Fe, since phases with its presence have been identified.
Respond: The EDS result of element Fe has been supplied in the revised manuscript.
Fig.9 SEM of the cross-section of Si and Si3N4 briquette (a) Si briquette; (b) Si3N4 briquette; EDS image of outer shell (c) Si briquette; (d) Si3N4 briquette

Reviewer 2 Report
The present manuscript is interesting, but the results are shown in such a way that it is closer to a good laboratory report rather than having significant scientific merit. I suggest improving the discussion since the change from amorphous to crystalline due to the high-temperature treatment is normal to occur; therefore, it is necessary to explain in more detail the reasons for the increase in the compressive strength of Si and Si3N4 briquette. It will be essential to mention the kind of organic binder since the question could be, what happens if I use another organic binder? There must be several organic binders.
Author Response
Response to the Reviewer’s Comments
We would like to thank you for your helpful comments. We have followed all the suggestions and made revisions accordingly. The list below gives answers to questions and the details of the changes that we have made in the revised manuscript.
Comments: The present manuscript is interesting, but the results are shown in such a way that it is closer to a good laboratory report rather than having significant scientific merit. I suggest improving the discussion since the change from amorphous to crystalline due to the high-temperature treatment is normal to occur; therefore, it is necessary to explain in more detail the reasons for the increase in the compressive strength of Si and Si3N4 briquette. It will be essential to mention the kind of organic binder since the question could be, what happens if I use another organic binder? There must be several organic binders.
Respond: The discussion of the change from amorphous to crystalline has been modified, and the reasons for the increase in the compressive strength of Si and Si3N4 briquettes are explained in detail in the manuscript as follows:
In the high-temperature process, both Si powder and Si3N4 powder on the briquette surface react with O2 and generate SiO2 easily as follows reactions (1) and (2):
Si + O2 → SiO2 (1)
Si3N4 + O2 → SiO2 + NO2 (2)
As shown in Fig.7, the SiO2 produced by the above reactions has undergone a process from amorphous to crystalline and makes the surface of briquettes more compact, and the compressive strength of the briquette improves with the growth of the shell thickness. Meanwhile, the growth of the outer shell stops since amorphous SiO2 blocks the pores on the surface of the briquette. Due to the particle sizes and shapes of Si particles being more uneven and irregular than that of Si3N4 particles, the size of pores on the surface of Si briquettes is too large and the O2 diffuses in the briquette more deeply. Thus, the outer shell of the Si briquette is thicker than that of the Si3N4 briquette, which conducts the compressive strength of the Si briquette improved more after high-temperature treatment.
Fig.7 Formation of SiO2 layer on the surface of briquettes after heating under an oxygen atmosphere
We are sorry that we could not provide more information about the organic binder in this manuscript because of some business reasons. But, the kinds of organic binders may affect the compressive strength of the briquette to a certain degree due to the bonding mechanisms of different binders being more different such as resin, honey, and so on. Thus, the choice of binder is an important part of cold-pressing. Meantime, a suitable additive amount also determines the cost and the compressive strength of the briquette.

Round 2
Reviewer 1 Report
The authors answered all the questions posed, the article can be accepted for publication.
This manuscript is a resubmission of an earlier submission. The following is a list of the peer review reports and author responses from that submission.